# Predictive Models for Nurses’ Entrepreneurial Intentions Using Comparison of Competing Models

**DOI:** 10.3390/ijerph19106027

**Published:** 2022-05-16

**Authors:** Ye-Jung Kim, Ji-Young Lim

**Affiliations:** 1Department of Nursing, Wooridul Hospital, Hanulgil 70, Gangseogu, Seoul 07505, Korea; cole2000@naver.com; 2Department of Nursing, Inha University, 100 Inharo Michuholgu, Incheon 22212, Korea

**Keywords:** entrepreneurship, intention, nurses, structural equation model, theory of planned behavior

## Abstract

There is a need to introduce predictive models of nurses’ entrepreneurial intentions that can identify which variables will promote entrepreneurship among nurses. This study aimed to explore the factors influencing nurses’ entrepreneurial intentions. We performed a systematic review and developed prediction models using factors identified and validated in a meta-analysis. Moreover, we individually tested and compared three models based on: (1) the Theory of Planned Behavior, (2) a meta-analysis, and (3) a combination of the two. Data from 386 nurses were analyzed using SPSS 23.0 for Windows and AMOS 21.0. The squared multiple correlation statistics of Models 1, 2, and 3 were 54.3%, 35.8%, and 60.0%, respectively. Model 3 provided a better explanation of nurses’ entrepreneurial intentions. Attitude, subjective norm, perceived behavioral control, entrepreneurial orientation, and need for entrepreneurship education are the most important variables to strengthen the entrepreneurial intention of nurses. The results of this study can be used as a theoretical model to explain nurse entrepreneurship intentions. In addition, these findings offer a useful resource for constructing a start-up curriculum within nursing colleges that fosters prospective nursing entrepreneurs.

## 1. Introduction

Entrepreneurial competence, which is the ability to obtain, organize, and utilize the resources required to start and grow a new organization, is essential for nurses and should be established as a skill for nursing management [1]. To adapt to shifting paradigms, nurses need to be capable of becoming leaders and managers, entrepreneurs, and employers on healthcare teams. The unique experience of nurses makes them one of the healthcare personnel who have the requisite competencies for entrepreneurship, as they can generate opportunities focused on health activities as healthcare experts, while also contributing innovative approaches and solutions to health problems in diverse social contexts [1].

Entrepreneurial intention is a prerequisite for nurse entrepreneurship. Recent studies have shown that entrepreneurial intention is predictive of long-term future entrepreneurial activities; based on these findings, Liñán and Fayolle [2] argued that entrepreneurial intention is the most important factor in entrepreneurial processes. Thus, understanding entrepreneurial intention is imperative to understand the entire entrepreneurial process [3]. In the field of nursing entrepreneurship in South Korea, despite the existence of legal standards and several entrepreneurial fields available for nurses, the number of nurse entrepreneurs is extremely small; thus, it is not captured by statistics. There has also been a shortage of research on nurse entrepreneurship, with only a few exploratory studies of entrepreneurial intention among nursing students [4,5,6].

Research on entrepreneurial intention can be broadly divided into two fields: research on entrepreneurship and analysis of entrepreneurial intention. The former aims to investigate personal characteristics linked to entrepreneurship, while the latter aims to reveal the psychological process by which attitudes and beliefs about entrepreneurship lead to effective entrepreneurial behavior [2]. Given that entrepreneurship models present different personal and environmental factors that affect entrepreneurial intention, there are limitations when selecting a specific model to explain entrepreneurial intention. In particular, there has been persistent criticism about the lack of theoretical basis for such a position, as an individual’s personality or environment is not a variable that can predict future entrepreneurship behaviors [7,8,9]. Conversely, entrepreneurial intention models provide practical insights into entrepreneurs’ planned and perceived behaviors and have been highlighted as a useful tool for predicting future entrepreneurial behavior [2]. Although the growth of nurse entrepreneurship requires environmental support and the development of nurses’ personal characteristics, there is a need to propose predictive models of nurses’ entrepreneurial intentions that can determine which attitudes and beliefs will lead to nurse entrepreneurship among experienced nurses.

To this end, in the present study, three models were constructed: Model 1 (comparative) applied the Theory of Planned Behavior of Ajzen [10]; Model 2 (comparative) was constructed from variables that were validated through a systematic literature review and meta-analysis of previous studies on entrepreneurial intention; and Model 3 (main) was constructed as a mixture of these two models. Model 3 was then compared with the other two competing models to develop a simple model that explains nurses’ entrepreneurial intentions.

The objective of the present study was to explore the structural relationships between factors affecting hospital nurses’ entrepreneurial intentions and compare competing models to develop a simple model with the strongest explanatory power.

## 2. Materials and Methods

### 2.1. Construction of the Structural Equation Model

This study aimed to develop and validate a simple model for exploring the structural relationships between factors affecting hospital nurses’ entrepreneurial intentions based on the conceptual framework of the Theory of Planned Behavior. Structural equation modeling is a multivariate analysis technique that facilitates inferences about the causal relationships between variables when it is difficult or impossible to conduct experimental research [11]. A structural equation model enables statistical evaluation of the theoretical model, making it possible to accept or modify the developed model as a valid one, by evaluating the fit of the theoretical model to the actual data [12]. When a model is nested in another model, it is possible to statistically verify which model is better using the chi-square values of the two models. A standard chi-square value of 2 or less is considered excellent [11]. Thus, if a hypothesis is rejected, which indicates that there is a significant difference between the two competing models, a model with fewer degrees of freedom is selected. However, if the hypothesis is not rejected, a simpler model—a model with more degrees of freedom—is selected. Therefore, in this study, Model 1 was not compared because it was a saturated model that was already verified. As Model 3 was nested in Model 2, Models 2 and 3 were compared and verified using the competitive model evaluation method.

### 2.2. Model 1

To construct Model 1 for explaining hospital nurses’ entrepreneurial intentions, Ajzen’s [10] Theory of Planned Behavior was used. The Theory of Planned Behavior claims that three main factors affect intention, and intention leads to actual behavior. These three factors are as follows: Individuals’ positive or negative attitudes toward the target behavior; subjective norms, which represent the influence of a reference population on the decision to perform the target behavior; and perceived behavior control, which is the individuals’ perception of how easy or difficult it is to perform the target behavior. The Theory of Planned Behavior has been widely supported as a theoretical model of entrepreneurial intention and has been validated as a strong predictor of entrepreneurial intention in various studies on office workers, construction technicians, senior entrepreneurs, deluxe hotel employees, potential female entrepreneurs, workers in the food service industry, and college students [13,14,15,16,17,18].

### 2.3. Model 2

To construct Model 2, a systematic literature review and meta-analysis of previous studies on entrepreneurial intention were performed. First, the inclusion criteria for studies in the systematic literature review were as follows: studies published in English or Korean that included, among the research variables, factors related to entrepreneurial intention and that provided correlation coefficients in the results. The exclusion criteria were as follows: studies that did not include factors related to entrepreneurial intention among the research variables, studies that did not report correlation coefficients, studies published in a language other than English or Korean, studies that only provided an abstract or only presented partial results at a conference, and studies for which the full text could not be obtained.

The literature search was performed using ScienceDirect, Medline, and Embase as international academic databases, and the Research Information Sharing System, the National Assembly Library, the Korean studies Information Service System, Kmbase, and the National Digital Science Library as domestic academic databases. Studies published between January 2009 and September 2019 were selected. Using the population, intervention, comparison, and outcome (PICO) system, P was set as nursing students and nurses, I as entrepreneurial intention, and C and O were not restricted.

The search results showed a total of 5832 studies, with 4495 studies from ScienceDirect, 13 studies from Embase, 411 studies from the Research Information Sharing System, 286 studies from the National Assembly Library, 83 studies from the Korean studies Information Service System, 1 study from Kmbase, and 535 studies from the National Digital Science Library. Among these, 636 duplicate studies were excluded. The titles and abstracts of the remaining 5196 studies were inspected, and 4985 studies unrelated to entrepreneurial intention were excluded. The full texts of the remaining 211 studies were inspected, of which 95 studies were removed based on the exclusion criteria, and 116 studies were selected. Finally, using the quality assessment tool for correlational studies developed by Cummings and Estabrooks [19], the quality of the selected studies was assessed, which led to the exclusion of 12 studies that scored eight points or less and the inclusion of the remaining 104 studies in the final analysis. The selection and exclusion of studies were independently conducted by the researcher and an independent evaluator with a Ph.D. in nursing management. Cohen’s kappa coefficient indicated good inter-rater reliability (κ = 0.90). Articles that did not conform to the inclusion criteria were discussed by the independent evaluator and the authors until a consensus was reached regarding their inclusion or exclusion (Figure 1).

Next, a meta-analysis using Comprehensive Meta-Analysis 3.0 was performed. To analyze the effect size of each entrepreneurial intention-related factor, the effect size r (ESr) was obtained based on the correlation coefficient, and the r statistic was converted to Fisher’s z. The mean weighted correlational coefficient was used to calculate the total effect size by assigning weights based on the number of cases. The criteria given by Cohen [20] were used to interpret the final effect size, where an ESr of 0.30 is categorized as a “moderate effect,” and after obtaining the 95% confidence interval for the calculated effect size, the statistical significance is determined by checking whether “0” is included in the interval. Funnel plots and fail-safe N were used to analyze publication bias, confirming the absence of publication bias (Figure 2).

In the results of the systematic literature review and meta-analysis, eight variables (self-efficacy, need for achievement, entrepreneurship, entrepreneurial orientation, social capital, funding, economic preparation, and need for entrepreneurial education) were significantly correlated with entrepreneurial intention (Table 1).

### 2.4. Model 3

Model 3, which explains hospital nurses’ entrepreneurial intentions, was constructed by combining Models 1 and 2. Here, the exogenous variables of Model 3 were the three components of the Theory of Planned Behavior, and the endogenous variables consisted of nine factors: eight variables extracted from the meta-analysis and entrepreneurial intention (Figure 3). The eight variables derived from the meta-analysis were classified into two broad groups, namely personal factors and environmental factors, according to the results of previous studies [21,22]. Personal factors included self-efficacy, need for achievement, entrepreneurship, and entrepreneurial orientation, while environmental factors comprised social capital, funding, economic preparation, and need for entrepreneurial education.

### 2.5. Sample

The study participants were nurses working at one of four general hospitals in three cities for at least one year. The participants voluntarily consented to participate in the study based on their understanding of the study objectives. A sample size of 400 persons was selected, with 100 persons from each of the four hospitals. This methodology was based on a study by Hoogland and Boomsma [23], who concluded that a sample size of *n* = 200–400 was required for comparing competing models using structural equations. Questionnaire responses were received from 396 persons (99.0%), of which 10 were excluded due to missing values, and the remaining 386 responses (96.5%) were included in the final analysis.

### 2.6. Measures

#### 2.6.1. Entrepreneurial Intention

Entrepreneurial intention was measured using an instrument developed by Crant [24] and later modified by Lee [25]. It comprises five questions, each scored on a five-point Likert scale, with higher scores indicating stronger entrepreneurial intention. Cronbach’s α was 0.935 in Lee’s study [25] and 0.925 in the present study.

#### 2.6.2. Planned Behavior

Planned behavior was measured using a version of the Entrepreneurial Intention Questionnaire developed by Liñán and Chen [26] and later modified by Sung [27]. It comprises 14 questions, with 5 questions on attitudes, 3 on subjective norms, and 6 on perceived behavioral control. Each question was rated on a five-point Likert scale, with higher scores indicating more positive attitudes, adherence to subjective norms, and perceived behavioral control. Cronbach’s α for each component was 0.897, 0.773, and 0.885, respectively, in Sung’s study [27], and 0.749, 0.930, and 0.792, respectively, in the present study.

#### 2.6.3. Self-Efficacy

Self-efficacy was measured using an instrument developed by Gong [28], based on Bandura’s [29] theory. It comprises seven questions, each scored on a five-point Likert scale, with higher scores indicating higher self-efficacy. Cronbach’s α was 0.853 in Gong’s study [28] and 0.868 in the present study.

#### 2.6.4. Need for Achievement

The need for achievement was measured using an instrument developed by Hwang [15]. It comprises nine questions, each rated on a five-point Likert scale, with higher scores indicating a higher need for achievement. Cronbach’s α was 0.773 in Hwang’s study [15] and 0.861 in the present study.

#### 2.6.5. Entrepreneurship

Entrepreneurship was measured using an instrument developed by Lee [30], based on an instrument previously used by Covin and Slevin [31]. It comprises 12 questions, including 4 questions on innovation, 4 on risk-taking, and 4 on proactiveness. Each question was rated on a five-point Likert scale, with higher scores indicating stronger entrepreneurship. Cronbach’s α for each component was 0.832, 0.802, and 0.747, respectively, in Lee‘s study [30], and 0.871, 0.778, and 0.782, respectively, in the present study.

#### 2.6.6. Entrepreneurial Orientation

Entrepreneurial orientation was measured using a version of the Career Orientation Inventory revised to 15 questions—3 questions on safety orientation, 3 on autonomy orientation, 3 on entrepreneurial orientation, 3 on technology orientation, and 3 on management orientation—by Yoon [22], based on the original instrument developed by Schein [32]. Each question was rated on a five-point Likert scale, with higher scores indicating stronger entrepreneurial orientation. Cronbach’s α was 0.876 in Yoon‘s study [22] and 0.837 in the present study.

#### 2.6.7. Social Capital

Social capital was measured using an instrument developed by Kim [33], based on a study by Yoon [22]. It comprises eight questions, each rated on a five-point Likert scale, with higher scores indicating greater awareness of social capital. Cronbach’s α was 0.841 in Kim‘s study [33] and 0.935 in the present study.

#### 2.6.8. Funding

Funding was measured using an instrument developed by Lee [34]. It comprises four questions, each rated on a five-point Likert scale, with higher scores indicating greater awareness of funding. Cronbach’s α was 0.813 in Lee‘s study [34] and 0.783 in the present study.

#### 2.6.9. Economic Preparation

Economic preparation was measured using an instrument developed by Kim [33]. It comprises five questions, each rated on a five-point Likert scale, with higher scores indicating a higher level of economic preparation. Cronbach’s α was 0.841 in Kim’s study [33] and 0.870 in the present study.

#### 2.6.10. Need for Entrepreneurship Education

The need for entrepreneurship education was measured using an instrument developed by Park [35], based on a study by Yook [21]. It comprises four questions, each rated on a five-point Likert scale, with higher scores indicating a stronger need for entrepreneurship education. Cronbach’s α was 0.841 in Park’s study [35] and 0.856 in the present study.

### 2.7. Data Collection and Ethical Considerations

Data for this study were collected between January 21st and February 20th, 2020. Before starting the study, the data collection methods were reviewed and approved by the Institutional Review Board of our hospital for the participants’ protection. The authors visited the nursing department of each hospital to explain the study objectives and requested cooperation. After obtaining consent according to the research approval processes of each hospital, the corresponding nursing units were visited, the study objectives and methods were explained, and the questionnaires were distributed.

Participants were informed that they could withdraw from the study at any time. The self-report questionnaires and consent forms were submitted in separate sealed envelopes to protect the participants’ anonymity. The authors retrieved the sealed questionnaire envelopes by visiting the relevant nursing units. For rural locations, the chief nurse of the unit delivered sealed envelopes to the nursing department, from which they were mailed to the authors.

To protect participants’ personal information, the responses were immediately coded and converted to numbers. Participants were provided with a small token of appreciation for participating in the study.

### 2.8. Data Analysis

SPSS version 23.0 (IBM SPSS software) and AMOS 21.0 (IBM AMOS) were used for statistical analysis. The raw values, percentages, means, and standard deviations were calculated to analyze the participants’ general characteristics and measured variables. Normality was tested using standardized skewness and kurtosis. The validity of the research instruments was tested using principal component analysis with varimax rotation. Reliability was tested by calculating Cronbach’s α, and correlations between the measured variables were tested by calculating Pearson’s correlation coefficients. The measurement model was estimated for structural equation modeling, and then the structural model was estimated. Confirmatory factor analysis was performed to assess the validity of the latent variables in the measurement model.

To test the research hypothesis, a structural equation model was constructed, fit indices were calculated to examine the fit between the paths and the data, and the explanatory power was analyzed. Model fit was analyzed using the badness-of-fit indices x2 and CMIN/df, the absolute fit indices goodness-of-fit index and adjusted goodness-of-fit index, and the relative fit indices comparative fit index, Tucker–Lewis index, normed fit index, incremental fit index, and root mean square error of approximation and standardized root mean square residual. Bootstrapping was used to calculate the indirect and total effects of the research model and to test their statistical significance. Finally, a competing model assessment was performed by comparing the indices of fit and explanatory power of Models 1, 2, and 3. Competing model assessment refers to the process of presenting several theoretically possible models and comparing the models to select a final model with high explanatory power, which is easy to interpret and fits the data appropriately [36].

## 3. Results

### 3.1. Participants’ General Characteristics

The general characteristics of the participants are shown in Table 2.

### 3.2. Descriptive Statistics and Confirmatory Factor Analysis of the Measured Variables

Descriptive statistics and confirmatory factor analysis of the measurement model results are shown in Table 3. Among personal factors, the need for achievement had the highest score, and of the environmental factors, economic preparation had the highest score.

The range of correlation coefficients for entrepreneurial intention with the measured variables indicated that they were all positive correlations. Moreover, the range of correlation coefficients between the measured variables indicated no problems due to multicollinearity. Among the research variables in this study, a confirmatory factor analysis demonstrated that the need for achievement, entrepreneurship, and social capital showed standardized factor loading ≥ 0.5, construct reliability (CR) ≥ 0.7, and average variance extracted (AVE) ≥ 0.5, thus evidencing the convergent validity of these three variables.

### 3.3. Testing the Fit of Model 3 (Main Model)

For Model 3, which was the main model used in this study, x^2^ was 109.09, CMIN/df was 2.32, and the absolute fit and relative fit indices were all ≥0.9. The parsimony-adjusted indices root mean square error of approximation and standardized root mean square residual were ≤0.5, indicating a good fit. The model was simplified by removing non-significant paths. The fit indices of the simplified model were as follows: x^2^ = 2.46, CMIN/df = 1.23, goodness-of-fit index = 0.998, adjusted goodness-of-fit index = 0.978, comparative fit index = 1, Tucker–Lewis index = 0.996, normed fit index = 0.997, incremental fit index = 1, root mean square error of approximation = 0.024, and standardized root mean square residual = 0.014.

### 3.4. Path Coefficients and Effects in Model 3 (Main Model)

The path coefficients analysis of the simplified Model 3 reported effects of subjective norms and perceived behavioral control on entrepreneurial orientation, with an explanatory power of 17.1%. Similarly, attitude and subjective norms affected the need for entrepreneurial education, with an explanatory power of 15.8%. Furthermore, attitude, subjective norms, perceived behavioral control, entrepreneurial orientation, and need for entrepreneurial education reported significant effects on entrepreneurial intention, with an explanatory power of 60.0% (Figure 4).

### 3.5. Comparison with Competing Models

When competing structural equation models are compared, the fit of each model needs to satisfy the required criteria. Model 1 was a saturated model with df = 0, x^2^ = 0, and goodness-of-fit index = 0, while the fit of Models 2 and 3 both satisfied the criteria. When the competing models were compared, Model 1 explained 54.3% of nurses’ entrepreneurial intentions, Model 2 explained 35.8%, and Model 3 explained 60.0%. Because Model 1 is a saturated mode, upon comparison with Models 2 and 3, the standard x^2^ of Model 2 was 4.74, and that of Model 3 was 1.23. The explanatory powers of Models 2 and 3 were 35.8% and 60.0%, respectively. Therefore, Model 3 best explained the entrepreneurial intention of hospital nurses (Table 4).

## 4. Discussion

In the comparison of competing models, a smaller x^2^ is better, and CMIN/df ≤ 2–3 is considered excellent [37]. When the comparison models show good fit, it is logical to use CMIN/df, the chi-squared statistic divided by the degrees of freedom, and SMC to compare the models [38]. Comparison of the competing models in the present study revealed that Model 3, the main model, was the best suited for explaining hospital nurses’ entrepreneurial intentions. The determining factors that affected entrepreneurial intention were attitude, subjective norms, and perceived behavior control from the Theory of Planned Behavior, and entrepreneurial orientation and need for entrepreneurial education from the systematic literature review and meta-analysis. These variables showed a strong explanatory power of 60.0% for hospital nurses’ entrepreneurial intentions. These findings are consistent with the results of numerous previous studies that have demonstrated that the Theory of Planned Behavior is a powerful theoretical framework for explaining entrepreneurial intention [16,39,40].

To establish the foundations for nurses pursuing entrepreneurship, direct support and specific relationship-related curricula should be coupled with efforts to exert positive changes on nurses’ attitudes toward their entrepreneurship. On a larger scale, efforts to exert positive changes should also be targeted toward attitudes, perceptions, and emotional factors regarding nurse entrepreneurship in society and reference populations. This is because the attitudes, subjective norms, and perceived behavior control constituting entrepreneurial intention take time to improve and can be developed gradually through sustained, specific entrepreneurial education and role modeling for nurse entrepreneurs. To achieve this, it will be necessary to develop entrepreneurial and start-up courses at the management business administration (MBA) level, to enable nursing students and graduate students to systematically improve their entrepreneurial competencies. In particular, to help improve perceptions of entrepreneurship and develop entrepreneurial competencies among experienced nurses, it will be necessary to improve entrepreneurial competencies in the short and long terms. In the short term, entrepreneurial adaptation programs can be developed and practical entrepreneurial education, including marketing, funding, finance, and accounting, can be supported to improve perceptions. In the long term, strategies such as the provision of professional consulting can be looked into, along with exploring other avenues that can enhance perceived behavioral control.

In the present study, entrepreneurial orientation had a positive effect on entrepreneurial intention. This supports the results of previous studies by Go [14]. In addition, entrepreneurial orientation was identified as a mediating variable between perceived behavioral control and entrepreneurial intention. Thus, entrepreneurial intention increases nurses’ intrinsic confidence in entrepreneurship and improves their sense of extrinsic control (i.e., the ability to seize entrepreneurial opportunities and maximize the use of available resources), ultimately acting as a factor that drives entrepreneurial practice and success.

Kim and Lim [4] reported that experienced entrepreneurs identify business opportunities through the awareness of problems in their organizations or societies. This suggests that hospital nurses could implement original ideas and explore new entrepreneurial opportunities based on their ample experience in nursing work and their academic background in nursing. Exploring and recognizing opportunities is crucial for successful entrepreneurship, and Muñoz et al. [41] reported that this ability to seize opportunities could be developed sufficiently through entrepreneurial education.

The need for entrepreneurial education also has a positive effect on entrepreneurial intention. Future nurse entrepreneurs lack both experience and entrepreneurial knowledge; thus, they need entrepreneurial education in areas such as exploring entrepreneurial ideas and opportunities, validity analysis, business planning, successful management techniques, marketing, finance, accounting, and law [42]. Sharp and Monsivais [43] conducted a qualitative study of 24 nurse practitioners in the United States and reported difficulties in nurse entrepreneurship in relation to the range of business practices, business skills, and role conflict, arguing for the need for financial support and education in law, regulations, conventions, strategic planning, leadership, and nursing center management. As it is dangerous to start a business based on only willpower, the importance of entrepreneurial education is frequently emphasized. Many studies have reported that a high level of entrepreneurial education is an effective strategy to convert entrepreneurial intention into entrepreneurial action [44]. Learning entrepreneurial experience within an organization, such as establishing new strategies while performing nursing work in the hospital, can enhance nurses’ entrepreneurial attitudes and perceived behavioral control, and when future nurse entrepreneurial opportunities arise, they can be recognized and converted into entrepreneurial behavior. This is because, rather than simply repeating the same nursing work, there is a process of innovation that leads to improved work or ideas for product development by identifying areas that could be changed or improved.

The significance of this study is as follows: First, from a theoretical perspective, the nurses’ entrepreneurial intention prediction model presented in this study can be used to develop a start-up program for new nurses who are required to play a new role as an entrepreneur in an aging society with a focus on health promotion. Second, in terms of research, as entrepreneurship intention has been studied using a wide variety of variables, there is a lack of consensus regarding a single model that adequately explains entrepreneurial intention. Thus, this study is significant in that it developed an optimized model to explain nursing entrepreneurship by selecting variables based on statistical significance through extensive systematic literature review and meta-correlation analysis. Last, from a practical viewpoint, the study results can be used as basic data for developing a nursing entrepreneurship standard curriculum for nursing students and creating a nursing entrepreneurship training program for prospective nurse entrepreneurs.

### Limitations

As this study was conducted on nurses working at general hospitals in Korea, one must be cautious in generalizing the study results. This is because the nurse’s entrepreneurial intention was a dynamic variable influenced by social contexts. Nevertheless, the model derived in this study was differentiated from previous studies in that it was built based on extensive literature reviews and meta-analyses. Therefore, it is expected that the study results might be used as a conceptual framework for future studies to develop an explanatory model for nurses’ entrepreneurial intentions in various cultures.

## 5. Conclusions

In this study, three models were constructed to explain hospital nurses’ entrepreneurial intentions, and these models were compared using a competing model analysis. The significance of this study is twofold: (1) Through a comprehensive review of factors shown to affect entrepreneurial intention in previous studies from different perspectives, this study presents the simplest model derived from these factors, and (2) this study reiterates that the widely-accredited Theory of Planned Behavior model is a valid theory for explaining entrepreneurial intention and adds to existing research by providing an effective theoretical framework for explaining hospital nurses’ entrepreneurial intentions. Finally, a simple and explanatory model (Model 3) for hospital nurses’ entrepreneurial intentions was developed and validated, meeting the purpose of this study.

Although self-efficacy, need for achievement, entrepreneurship, and social capital have been suggested as factors affecting entrepreneurial intention in several previous studies, they were not significant in the present study. This could be an indirect effect of the sample population, which consisted of hospital nurses with stable employment; therefore, further studies are needed to investigate the cause of these discrepancies. In addition, there is a need for international research comparing nurses’ entrepreneurial intentions, entrepreneurial competencies, and the actual state of entrepreneurial activities between countries to analyze the institutions and policies in countries with the most active nurse entrepreneurship and help direct benchmarking efforts. Given the positive effect of entrepreneurial education on nurses’ entrepreneurial intentions, developing multidimensional programs to foster entrepreneurship and support nurses’ entrepreneurial activities is essential.

## Figures and Tables

**Figure 1 ijerph-19-06027-f001:**
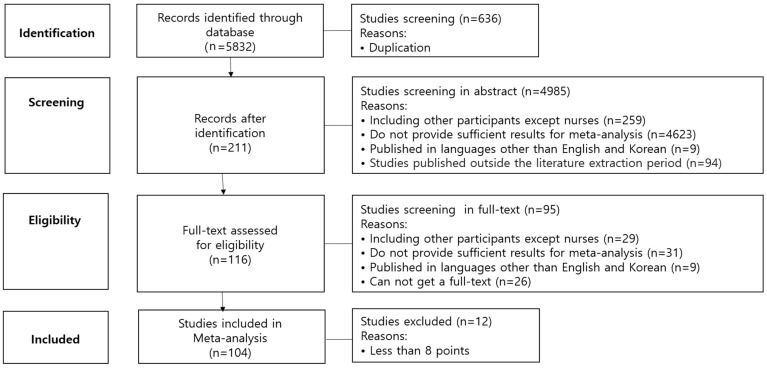
Flow diagram of study selection.

**Figure 2 ijerph-19-06027-f002:**
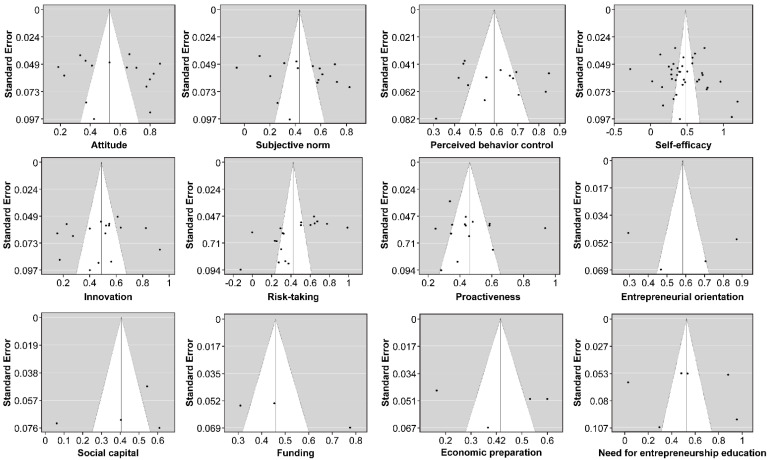
Summary of funnel plots.

**Figure 3 ijerph-19-06027-f003:**
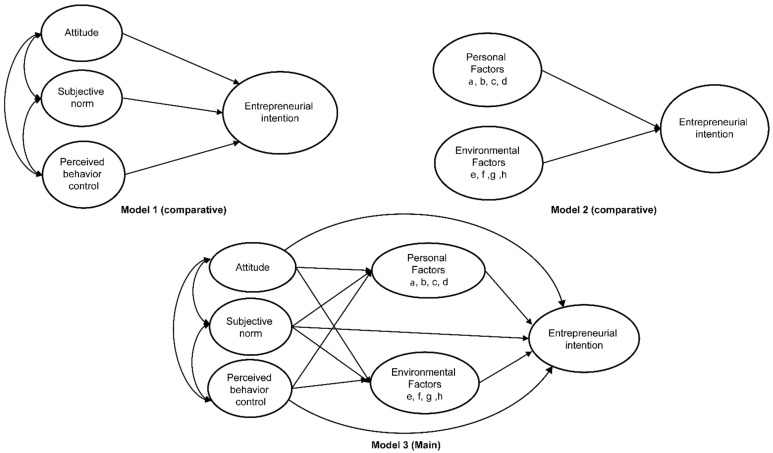
Hypothetical models of this study. (**a**) Self-efficacy. (**b**) Need for achievement. (**c**) Entrepreneurship. (**d**) Entrepreneurial orientation. (**e**) Social capital. (**f**) Funding. (**g**) Economic preparation. (**h**) Need for entrepreneurship education.

**Figure 4 ijerph-19-06027-f004:**
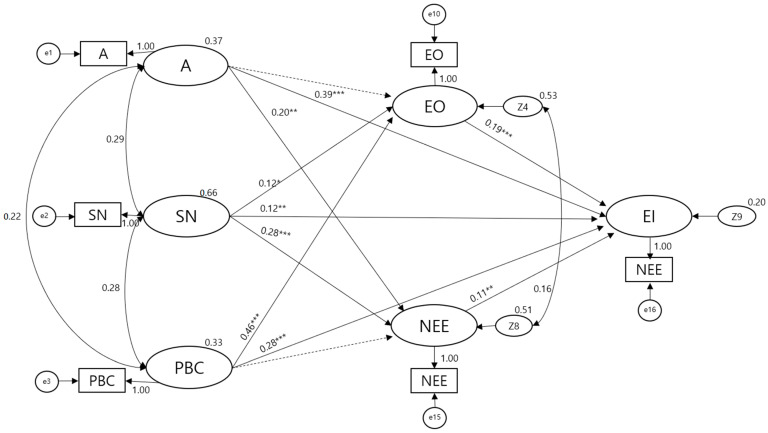
Modified Model 3 of this study. A, attitude; SN, subjective norm; PBC, perceived behavior control; EO, entrepreneurial orientation; NEE, need for entrepreneurship education; EI, entrepreneurial intention. *** *p* < 0.001, ** *p* < 0.01, * *p* < 0.05.

**Table 1 ijerph-19-06027-t001:** Summary of meta correlation analysis.

Variables		*n*	Effect Size r	Lower Limit	Upper Limit	I^2^ (%)	Q	*p*	Tau^2^
Theory of planned behavior	Attitude	17	0.492	0.404	0.571	3.00	16.49	<0.001	0.85
	Subjective norm	17	0.408	0.306	0.500	0.00	15.34	<0.001	0.96
	Perceived behavior control	14	0.529	0.469	0.584	6.49	13.90	<0.001	0.26
Personal factors	Self-efficacy	39	0.442	0.373	0.506	21.11	48.17	<0.001	2.56
	Need for achievement	13	0.353	0.282	0.420	0.00	11.33	<0.001	0.22
	Innovation	17	0.452	0.374	0.525	8.53	17.49	<0.001	0.61
	Risk-taking	21	0.402	0.309	0.488	3.75	20.78	<0.001	1.21
	Proactiveness	15	0.435	0.365	0.500	3.33	14.48	<0.001	0.36
	Entrepreneurial orientation	4	0.526	0.293	0.700	0.00	2.38	<0.001	0.32
Environmental factors	Social capital	4	0.385	0.180	0.558	10.83	3.37	<0.001	0.19
	Funding	4	0.429	0.260	0.572	18.31	3.67	<0.001	0.14
	Economic preparation	4	0.393	0.203	0.555	0.00	2.49	<0.001	0.17
	Need for entrepreneurship education	6	0.487	0.258	0.665	8.84	5.49	<0.001	0.64

**Table 2 ijerph-19-06027-t002:** General characteristics (*n* = 386).

Variables	Categories	*n*	%	M ± SD
Gender	female	363	94.0	
	male	23	6.0	
Age	≤29	136	35.2	33.98 ± 7.13
	30~39	151	39.2	
	40~49	90	23.3	
	≥50	9	2.3	
Education	College	152	39.4	
	University	216	56.0	
	Graduate	18	4.6	
Marital status	Married	173	44.8	
	Unmarried	209	54.2	
	Others	4	1.0	
Department	Ward	220	57.0	
	Operation room	105	27.2	
	Out-patient department	51	13.2	
	Others	10	2.6	
Position	Staff nurse	256	66.3	
	Charge nurse	84	21.8	
	Head nurse	46	11.9	
Family start-up experience	Yes	152	39.4	
	No	234	60.6	
Whether to take EE	Yes	9	2.3	
	No	377	97.7	
Desired nursing start-up field *	Elderly care facility	144	37.3	
	Alternative nursing	59	15.3	
	Medical institution consulting	45	11.7	
	Postpartum care and breastfeeding	40	10.4	
	Nursing-related academy	32	8.3	
	Kindergarten	26	6.7	
	Lunch box for chronic disease	24	6.2	
	Rental shop for elderly equipment	16	4.2	
	Hospice center	16	4.2	
	Others	37	9.6	
Hindering factors for nursing start-up *	Lack of funds	180	46.6	
	Lack of knowledge	151	39.1	
	Fear of failure	65	16.8	
	Lack of confidence	63	16.3	
	Worries around	22	5.7	
	Others	13	3.4	
Success factors for nursing start-up *	Start-up knowledge	159	41.2	
	Start-up fund	125	32.4	
	Start-up role model	98	25.4	
	Start-up experience	49	12.7	
	Start-up confidence	25	6.5	
	Others	6	1.6	

EE = Entrepreneurship education; * multiple responses available.

**Table 3 ijerph-19-06027-t003:** Descriptive statistics and confirmatory factor analysis of measurement model.

Variable	Range	M ± SD	Skewness	Kurtosis	Standardized Factor Loading	AVE	CR
Attitude	1–5	2.78 ± 0.74	−1.97	1.47			
SN	1–5	2.92 ± 0.81	−3.09	0.08			
PBC	1–5	2.60 ± 0.67	−2.80	1.09			
Self-efficacy	1–5	3.38 ± 0.56	−0.90	−0.09			
Need for achievement		3.41 ± 0.54	−0.77	1.07		0.871	0.930
Relative NA	1–5	3.36 ± 0.58	−0.65	0.87	0.76		
Active NA	1–5	3.53 ± 0.61	−0.06	0.40	0.92		
Entrepreneurship		3.06 ± 0.51	2.93	3.16		0.692	0.865
Innovation	1–5	3.00 ± 0.58	2.32	1.04	0.77		
Risk-taking	1–5	3.35 ± 0.61	−0.05	1.34	0.81		
Proactiveness	1–5	3.22 ± 0.72	2.58	0.58	0.47		
EO	1–5	2.76 ± 0.81	2.07	0.65			
Social capital		2.65 ± 0.58	3.25	0.42		0.824	0.904
GEN	1–5	2.83 ± 0.92	1.52	0.00	0.88		
NEN	1–5	2.37 ± 0.83	3.02	0.39	0.86		
Funding	1–5	3.00 ± 0.78	−1.71	1.26			
EP	1–5	3.24 ± 0.72	−0.52	0.13			
Need for EE	1–5	3.14 ± 0.79	−1.41	3.27			
EI	1–5	2.49 ± 0.71	0.25	−0.88			

SN = subjective norm; PBC = perceived behavior control; Relative NA = relative need for achievement; Active NA = active need for achievement; EO = entrepreneurial orientation; GEN = general entrepreneurship network; NEN = nursing entrepreneurship network; EP = economic preparation; Need for EE = need for entrepreneurship education; EI = entrepreneurial intention.

**Table 4 ijerph-19-06027-t004:** Fitness of models.

Model	Endogenous Variables	Exogenous Variables	S.E	C.R (*p*)	Direct Effect (*p*)	Indirect Effect (*p*)	Total Effect (*p*)	SMC
M1	EI	Attitude	0.05	7.85 (<0.001)	0.427 (<0.001)		0.427 (<0.001)	0.543
		SN	0.07	6.29 (<0.001)	0.269 (<0.001)		0.269 (<0.001)	
		PBC	0.05	4.03 (<0.001)	0.309 (<0.001)		0.309 (<0.001)	
M2	EI	EO	0.05	5.36 (<0.001)	0.252 (<0.001)		0.252 (<0.001)	0.358
		Social capital	0.07	3.98 (<0.001)	0.274 (<0.001)		0.274 (<0.001)	
		Need for EE	0.04	5.11 (<0.001)	0.214 (<0.001)		0.214 (<0.001)	
M3	EI	Attitude	0.05	7.58 (<0.001)	0.334 (<0.001)	0.021 (0.010)	0.355 (<0.001)	0.600
		SN	0.04	3.13 (0.002)	0.137 (0.002)	0.052 (0.001)	0.189 (0.002)	
		PBC	0.06	4.99 (<0.001)	0.229 (<0.001)	0.086 (<0.001)	0.315 (<0.001)	
		EO	0.03	5.73 (<0.001)	0.214 (<0.001)		0.214 (<0.001)	
		Need for EE	0.03	3.23 (0.001)	0.120 (0.001)		0.120 (0.001)	

EI = entrepreneurial intention; SN = subjective norm; PBC = perceived behavior control; EO = entrepreneurial orientation; Need for EE = need for entrepreneurship education.

## Data Availability

The data presented in this study are available upon request from the corresponding author. They are not publicly available due to privacy and ethical restrictions.

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
