# Peer review of "Predictive Models for Nurses’ Entrepreneurial Intentions Using Comparison of Competing Models"

_ijerph, 2022, doi:10.3390/ijerph19106027_

Round 1
Reviewer 1 Report
Thank you very much for giving me the opportunity to review this paper. The aim of the article is to explore the factors influencing nurses’ entrepreneurial intentions. In my understanding, this is a very promising manuscript. Certainly, the authors have made significant discussion about its impact and made a significant contribution to relevant academic and practitioner literature. The theoretical foundation of the paper could be more solid. It will be better to add more recent references to support the discussion.
You have applied appropriate methodology and the results are also presented in an appropriate manner. You have highlighted the implications of this research at the end. It will be better to see the managerial and theoretical implications of this study as separate paragraph.
Please add a couple of references in the first paragraph of your article.
Please rewrite following sentence-
‘Based on the results of recent studies showing that entrepreneurial intention is predictive of long-term future entrepreneurial activities, Liñán and Fayolle [2] argued that entrepreneurial intention is the most important factor throughout the process of entrepreneurial behavior.’
Please use two sentences for the following statement-
Research on entrepreneurial intention can broadly be divided into two fields: research on entrepreneurship, which aims to investigate personal characteristics linked to entrepreneurship, and analysis of entrepreneurial intention, which aims to reveal the psychological processes by which attitudes and beliefs about entrepreneurship lead to effective entrepreneurial behaviour.
Author Response
Thank you very much for giving me the opportunity to review this paper. The aim of the article is to explore the factors influencing nurses’ entrepreneurial intentions. In my understanding, this is a very promising manuscript. Certainly, the authors have made significant discussion about its impact and made a significant contribution to relevant academic and practitioner literature. The theoretical foundation of the paper could be more solid. It will be better to add more recent references to support the discussion.
You have applied appropriate methodology and the results are also presented in an appropriate manner. You have highlighted the implications of this research at the end. It will be better to see the managerial and theoretical implications of this study as separate paragraph.
=> According to your comments, a paragraph explaining the methodology of the study was added in the Materials and Methods section, and the implications have been presented in a separate paragraph in the Discussion section. The changes are indicated in red font in the revised manuscript.
Please add a couple of references in the first paragraph of your article.
=> According to your comment, relevant references were added as needed.
Please rewrite following sentence-
‘Based on the results of recent studies showing that entrepreneurial intention is predictive of long-term future entrepreneurial activities, Liñán and Fayolle [2] argued that entrepreneurial intention is the most important factor throughout the process of entrepreneurial behavior.’
=> According to your comment, this sentence was rewritten in the manuscript.
Please use two sentences for the following statement-
Research on entrepreneurial intention can broadly be divided into two fields: research on entrepreneurship, which aims to investigate personal characteristics linked to entrepreneurship, and analysis of entrepreneurial intention, which aims to reveal the psychological processes by which attitudes and beliefs about entrepreneurship lead to effective entrepreneurial behaviour.
=> According to your comment, this sentence was revised as two separate sentences in the revised manuscript.
Thank you for your constructive comments

Reviewer 2 Report
Thank you very much for the opportunity to review the manuscript titled Predictive models for nurses’ entrepreneurial intentions using a comparison of competing models.
Entrepreneurship in the profession of nursing, entrepreneurial nursing, is seen as a prerequisite to handling global healthcare challenges, improving patient safety, and achieving excellence in nursing. The topic is very actual.
The meta-analysis raises my doubts. There is no consideration of the gold standard -randomized controlled experiments. The description of the methodology is weak. The conclusion does not provide information about achieving the objective of the study
Author Response
Thank you very much for the opportunity to review the manuscript titled Predictive models for nurses’ entrepreneurial intentions using a comparison of competing models.
Entrepreneurship in the profession of nursing, entrepreneurial nursing, is seen as a prerequisite to handling global healthcare challenges, improving patient safety, and achieving excellence in nursing. The topic is very actual.
The meta-analysis raises my doubts. There is no consideration of the gold standard -randomized controlled experiments.
=> Thank you for your valuable comments. In this study, meta-correlation analysis was performed based on survey studies. Therefore, the gold standard randomized controlled experiments were not considered because experiments or intervention studies were not subject to analysis.
The description of the methodology is weak.
=> According to your comment, a paragraph explaining the methodology of the study was added in the Materials and Methods section.
The conclusion does not provide information about achieving the objective of the study
=> According to your comment, a relevant sentence was added in the Conclusion section.
Thank you very much.
Reviewer 3 Report
The study, which aimed to explore the factors influencing nurses’ entrepreneurial intentions, is of great interest to the field of nursing. It is well written, and the English language and style are acceptable. Therefore, only minor changes are suggested. The research design is appropriate, and the manuscript is a condensed form of the first author’s doctoral dissertation from Inha University. The methods are adequately described, and the results are presented clearly. The four figures and the four tables help to summarise to convey the information.
According to Grammarly's suggestions, some English improvements are proposed (https://app.grammarly.com):
Consider replacing Kmbase with Embase. See attached document for additional suggestions.

Author Response
The study, which aimed to explore the factors influencing nurses’ entrepreneurial intentions, is of great interest to the field of nursing. It is well written, and the English language and style are acceptable. Therefore, only minor changes are suggested. The research design is appropriate, and the manuscript is a condensed form of the first author’s doctoral dissertation from Inha University. The methods are adequately described, and the results are presented clearly. The four figures and the four tables help to summarise to convey the information.
=> Thank you for your valuable comments.
According to Grammarly's suggestions, some English improvements are proposed (https://app.grammarly.com):
See attached document for additional suggestions.
=> According to your comment, this manuscript has been reviewed by a professional English editor based on the Grammarly report.
Consider replacing Kmbase with Embase.
=> Kmbase is a medical professional academic database used in Korea.
Thank you.
